# Biotransformation of Phytosterols into Androstenedione—A Technological Prospecting Study

**DOI:** 10.3390/molecules27103164

**Published:** 2022-05-15

**Authors:** Victor Oliveira Nunes, Nathália de Castro Vanzellotti, Jully Lacerda Fraga, Fernando Luiz Pellegrini Pessoa, Tatiana Felix Ferreira, Priscilla Filomena Fonseca Amaral

**Affiliations:** 1By&Bio—By-Products to Bioproducts Lab, Escola de Química, Universidade Federal do Rio de Janeiro, Rio de Janeiro 21941-909, RJ, Brazil; victorn@eq.ufrj.br (V.O.N.); nathalia_vanzellotti@hotmail.com (N.d.C.V.); jully.lfraga@gmail.com (J.L.F.); pessoa@eq.ufrj.br (F.L.P.P.); tatiana@eq.ufrj.br (T.F.F.); 2Centro Universitário SENAI CIMATEC, Salvador 41650-010, BA, Brazil

**Keywords:** steroid, androstenedione, biosynthesis, *Mycobacterium*, phytosterol, pharmaceutical

## Abstract

Androstenedione (AD) is a key intermediate in the body’s steroid metabolism, used as a precursor for several steroid substances, such as testosterone, estradiol, ethinyl estradiol, testolactone, progesterone, cortisone, cortisol, prednisone, and prednisolone. The world market for AD and ADD (androstadienedione) exceeds 1000 tons per year, which stimulates the pharmaceutical industry’s search for newer and cheaper raw materials to produce steroidal compounds. In light of this interest, we aimed to investigate the progress of AD biosynthesis from phytosterols by prospecting scientific articles (Scopus, Web of Science, and Google Scholar databases) and patents (USPTO database). A wide variety of articles and patents involving AD and phytosterol were found in the last few decades, resulting in 108 relevant articles (from January 2000 to December 2021) and 23 patents of interest (from January 1976 to December 2021). The separation of these documents into macro, meso, and micro categories revealed that most studies (articles) are performed in China (54.8%) and in universities (76%), while patents are mostly granted to United States companies. It also highlights the fact that AD production studies are focused on “process improvement” techniques and on possible modifications of the “microorganism” involved in biosynthesis (64 and 62 documents, respectively). The most-reported “process improvement” technique is “chemical addition” (40%), which means that the addition of solvents, surfactants, cofactors, inducers, ionic liquids, etc., can significantly increase AD production. Microbial genetic modifications stand out in the “microorganism” category because this strategy improves AD yield considerably. These documents also revealed the main aspects of AD and ADD biosynthesis: *Mycolicibacterium* sp. (basonym: *Mycobacterium* sp.) (40%) and *Mycolicibacterium neoaurum* (known previously as *Mycobacterium neoaurum*) (32%) are the most recurrent species studied. Microbial incubation temperatures can vary from 29 °C to 37 °C; incubation can last from 72 h to 14 days; the mixture is agitated at 140 to 220 rpm; vegetable oils, mainly soybean, can be used as the source of a mixture of phytosterols. In general, the results obtained in the present technological prospecting study are fundamental to mapping the possibilities of AD biosynthesis process optimization, as well as to identifying emerging technologies and methodologies in this scenario.

## 1. Introduction

Androstenedione (4-androstenedione or androst-4-ene-3,17-dione, CAS No.: 63-05-8), also known as 4-AD or only AD, is an androgen produced by the human body in the adrenal glands, as well as in the testicles and ovaries. Its importance is related to the human body’s ability to convert androstenedione into other hormones, such as testosterone and estrogen [1].

The AD derivative in the human body is produced from cholesterol. It can also be produced from phytosterols outside the body by a chemical or biochemical route [1]. In the biochemical route, first, the 3-hydroxyl-5-ene moiety of sterols is oxidized by 3β-hydroxysteroid dehydrogenase [2]. Then, two independent routes may be followed: steroid nucleus decomposition and side-chain cleavage [3], as shown in Figure 1. The enzymes 3-ketosteroid-9α-hydroxylase (Ksh) and 3-ketosteroid-1-dehydrogenase (KstD) catalyze the decomposition of the steroid nucleus, and, therefore, must be inactivated to increase AD production [4]. In the case of sitosterol, which is one of the most abundant phytosterols in vegetable oils, its branched side-chain cleavage begins at the nonpolar end of the sitosterol molecule. Then, a hydroxyl group is added to the C27 terminal methyl group, which is then oxidized to form a carbonyl. Subsequently, C28 carbon carboxylation occurs. Such steps are catalyzed by enzymes and are induced by the very presence of sitosterol [5].

According to Rokade et al. [1], there are limitations in the use of phytosterols as a substrate to produce AD, such as the low solubility of phytosterols in water and the degradation of the steroid core, which can lead to androstadienedione (ADD) or 9-hydroxy-androstenedione (9-OHAD) formation. Thus, the development of new techniques and biotechnological advances is important to increase AD productivity from phytosterols.

Technological prospecting studies provide information that leads to the identification of promising technologies, supporting the decision-making process. The systematization of the technological monitoring practice aims to bring together the search for appropriate solutions for the identification and the prioritization of a research and development (R&D) agenda, articulated with research institutions, which can even influence the national R&D agenda and create demand for the sector’s innovative chain [7,8].

Several comprehensive reviews, related to phytosterol conversion to steroids, that report an in-depth analysis of this process can be found in the literature, such as the ones published by Rokade [1] and Zhao et al. [9]. However, the decision to invest in a particular product, process, or segment, not only by companies but also by scientists who need to decide which research topic to follow, should be based on quantitative data related to the developed technology. For example, where this process is mainly studied (which country or institution), which are the major players—scientists or companies—of the scientific studies or patents, which are the most used or even overused strategies, what chemicals have already been tested, and so on. Therefore, a qualitative analysis of scientific articles and patents was adopted herein, and these data are not present in those reviews or any other document in the literature.

Therefore, the present work aims to identify and quantify data related to the scientific publications and patents that report the process of microbial AD production from phytosterols using the technological prospecting methodology to help scientists and companies in the decision-making process regarding how to invest their time and capital in this biotechnological process.

## 2. Materials and Methods

### 2.1. Systematic Search of Articles

A search in the Scopus, Web of Science, and Google Scholar databases was performed to systematically evaluate the progress in AD biosynthesis from phytosterols. For Scopus and Web of Science databases, in the prospective phase, the search was carried out from the combinations of keywords, filtering only those documents in which such combinations appeared only in the title, abstract, and keywords, as follows: “biotransformation” AND “4-androstenedione”; “biosynthesis” AND “4-androstenedione”; “biosynthesis” AND “androst-4-ene-3,17-dione”; “biotransformation” AND “androst-4-ene-3,17-dione”; “phytosterol” AND “androst-4-ene-3,17-dione”; “androstenedione” AND “microbial” AND “biotransformation”. For the Google Scholar database, as the search cannot be limited to title, abstract, and keywords—only to the title—a search without restriction was performed, with the terms always present in the relevant articles of the other bases searched: “androst” AND “biotransformation.” Without restricting the search period, 365 documents were obtained in Scopus, 41 papers in Web of Science, and 1940 documents in Google Scholar. A broad overview could reduce the Google Scholar search to 105 papers, excluding studies related to animal cells, human organs, reviews, and other products rather than AD, ADD, and similar metabolites. These searches were combined, and the duplicated documents were excluded.

After the dynamic reading of these documents (especially the abstracts), many non-relevant documents were identified, based on the central theme of AD biosynthesis from phytosterols. Studies that did not contain phytosterols (or cholesterol) as a substrate or AD, ADD, and similar metabolites as products were excluded. Therefore, a new search strategy was carried out, restricting the search to the period from January 2000 to December 2021, since many relevant documents fell within that period. As a result, 108 articles were obtained [4,6,10,11,12,13,14,15,16,17,18,19,20,21,22,23,24,25,26,27,28,29,30,31,32,33,34,35,36,37,38,39,40,41,42,43,44,45,46,47,48,49,50,51,52,53,54,55,56,57,58,59,60,61,62,63,64,65,66,67,68,69,70,71,72,73,74,75,76,77,78,79,80,81,82,83,84,85,86,87,88,89,90,91,92,93,94,95,96,97,98,99,100,101,102,103,104,105,106,107,108,109,110,111,112,113,114,115].

These 108 scientific articles were grouped into macro, meso, and micro perspectives. The macro perspective consisted in analyzing the year in which it was published, the type of institutions that published them (university, company, research center), and the country where these institutions are. Regarding the meso perspective, these publications were categorized according to their approach into four groups: “microorganism”; “process improvement”; “metabolic intermediates”, and “analytical methods and others”. The meso and micro categories are presented in Table 1.

The “microorganism” group gathered manuscripts that investigated the microbial producers of AD. Therefore, they were subcategorized as “genetic modification or genetic identification”—studies that genetically modified the microorganism, or used genetic tools to identify genes; “ks enzyme”—studies that were related to the main enzymes, Ksh and KstD, which regulate the metabolism of AD; “resting cells, cell-wall modifications or immobilization”—articles that used resting cells to produce AD, or that permeabilized the cell wall of the microorganisms, or immobilized them; and “microbial selection”—studies that selected the best AD producer, in the micro perspective.

Manuscripts related to the increase in AD production by modifying the bioprocess were categorized as “process improvement”. In the micro perspective, these works were subcategorized as “chemical addition”—studies related to the addition of β-cyclodextrins, inhibitors, solvents, polymers, etc., to increase AD production; “culture medium”—investigations that modified culture medium composition or substrate type (type of phytosterol) or substrate ratio to increase AD production; “biphasic system”—research performed to solubilize phytosterol and restrict solvent toxicity by the use of two-phase systems; “operational mode or strategy”—manuscripts describing the use of different bioreactor operation modes (fed-batch, sequential batch, etc.); and “process variables”—studies that evaluate the effect of certain process conditions, such as the agitation speed, the age and amount of inoculum, the temperature and oxygen level, etc.

The group “metabolic intermediates and hormones” combined studies that also report the production of other intermediates, such as ADD or 9-OHAD, as by-products or main products, from phytosterols (PS) or cholesterol (Co); however, they are of interest because of the similarity in the production process (subcategorized as “PS or Co conversion into intermediates”), as well as studies related to hormone production, as with testosterone, from phytosterols using the same type of process as AD production (subcategorized as “hormone production from PS”).

Some studies related to the analytical methodology for identifying substrates and products in the AD production process were categorized as “analytical methods and others”.

### 2.2. Systematic Search of Patents

The patent search methodology consisted of searching patents about AD in the USPTO (the United States Patent and Trademark Office) database [116], using different keyword combinations. The USPTO’s patent databases contain more than 10 million patents from 1976 to the present, with 5 million published applications from 2001 onward.

The keyword combinations used in the patent search were: “androstenedione AND (biotransformation OR biosynthesis)” and “androst-4-ene-3,17-dione AND (biotransformation OR biosynthesis)”. The USPTO Patent Full-Text and Image Database (PatFT) was used for a granted patents search, resulting in 682 documents in a timeframe from 1976 to December 2021. The USPTO Application Full-Text and Image Database (AppFT) was used for the patent application search, resulting in 837 documents from 2001 to December 2021.

Then, all documents were analyzed (title and abstract) and only those documents involving the biotransformation or biosynthesis of AD were selected for this study, resulting in 20 granted patents and 3 patent applications. The selected documents were also analyzed from three perspectives: macro, meso, and micro.

The macro perspective consisted of analyzing the year in which the patents were granted or filed; the types of patent-holding institutions (university, company, research center); and the country where those patent holders are based. Regarding the meso perspective, patents were categorized according to their approach into two groups: “technologies/routes” and “use of the molecule”. The patents categorized as “technologies/routes” were detailed from a micro perspective.

## 3. Results

### 3.1. Articles’ General Aspects—Macro Analysis

The number of publications fluctuated over the years, as Figure 2 shows, with a peak of 15 articles published in the year 2017. “Transparency Market Research” [117] reported that the fertility control rate dropped by 16% in the United States during the year 2017, leading to the expansion of the androstenedione market. This is probably why research on this topic increased so much in this period.

Figure 3 shows the distribution of manuscripts according to the country where the study was performed. China, the leading country, published 63 scientific articles, followed by India, Russia, Spain, and the EUA, which together published less than half of that (28 articles). In terms of the partnerships between countries, only 10 documents were detected: China/EUA (3), China/EUA/England (1), Portugal/Bulgaria (1); Russia/Vietnam (1); Russia/Spain (1), Spain/England (1) and India/Canada (1).

Most of the published scientific articles (76%) are studies developed in universities, followed by research centers/institutes (20%) and collaborations between universities and research centers (4%), according to Figure 4. Regarding Chinese publications, the Tianjin University of Science and Technology and the East China University of Science and Technology published the most scientific articles related to AD biosynthesis. The Russian Academy of Sciences is present in all publications from Russia, while Devi Ahilya University appears in the publications from India. The companies appear in scientific articles in collaboration with universities or research centers, as with the Russian Academy of Sciences. Pharmins, Ltd. (Moscow, Russia) is an independent private company founded in 2001, having its roots in the Laboratory of Microbial Transformation of Organic Compounds in the Institute of Biochemistry and Physiology of Microorganisms, Russian Academy of Sciences.

### 3.2. Categorizing the Articles by Groups—Meso Analysis

Four different categories were identified in the set of manuscripts found in the systematic search, these being “microorganism”, “process improvement”, “metabolic intermediates and hormones”, and “analytical methods and others” as shown in Figure 5. The study of the “process improvement” used in this bioprocess and the “microorganism” used for AD production are the themes most intensively studied (64 and 62 documents, respectively). The “metabolic intermediates and hormones” have also been significantly studied (37 publications). Only 5 manuscripts were categorized as “analytical methods and others”.

*Mycobacterium* sp. and *Mycobacterium neoaurum* are the most recurrent species studied, accounting for 40% and 32% of the studied species, respectively. In 2018, Gupta et al. [118] proposed the division of the genus *Mycobacterium* into 5 genera, including the genus *Mycobacterium,* comprising all the major human pathogens of this genus, and four novel genera, viz. *Mycolicibacterium*, *Mycolicibacter*, *Mycolicibacillus,* and *Mycobacteroides*. *Mycolicibacterium* corresponds to the *Fortuitum–Vaccae* clade, which includes the soil-origin species that produce AD [9,118,119]. Therefore, in the present study, the new classification will be used to refer to these species, even if the cited manuscript reports the original classification. Several genetic engineering strategies were used for *Mycolicibacterium neoaurum* (basonym: *Mycobacterium neoaurum*), to increase AD production, as in the deletion of two genes in *Mycolicibacterium smegmatis* mc²155 to enable phytosterol assimilation instead of just cholesterol and AD production [6].

Documents from the “process improvement” category addressed the increase in phytosterol bioconversion to AD by increasing steroid solubility [18,38,44,48,49,52,78,103,106,115], using several substances (D,L-norleucine, m-fluorophenialanine [106], ionic liquids [38,52,114], surfactants [40,78,103,112], and organic solvents [18,44,48,49,78]). This category also addresses the process variables involved in AD production, such as oxygenation control [59], agitation [23], temperature [54,64,112], substrate concentration [22,36,54,80,112], etc. Production strategies, such as the use of a two-phase system [23,27,66,68,99], are mentioned in several works. Table 2 summarizes the main results related to the analysis of scientific articles and patents.

#### 3.2.1. Scientific Articles Categorized as “Microorganism”—Micro Analysis

Of the 62 documents prospected in the “microorganism” category, which gathers papers in which the main target to increase AD production is the microorganism, 15 articles describe the use of *Mycolicibacterium* sp. (mentioned as *Mycobacterium* sp.) (for example: [19,21,24,32,110]), while 17 describe the use of *Mycolicibacterium neoaurum* (mentioned as *Mycobacterium neoaurum*) (examples: [10,11,13,14,30]). The remaining articles cited the following species as possible AD producers: *Mycolicibacterium smegmatis* (mentioned as *Mycobacterium smegmatis*) [6,47,95], *Mycolicibacterium fortuitum* (mentioned as *Mycobacterium fortuitum*) [18,37], *Nocardia* sp. [22], *Mycolicibacterium vaccae* (mentioned as *Mycobacterium vaccae*) [106,107], *Micrococcus roseus* [96], *Bacillus subtilis* [53], *Fusarium moniliforme* [70], *Moraxella ovis* [71], *Aspergillus oryzae* [89], *Aspergillus nidulans* [95], *Alkalibacterium olivoapovliticus* [109], and *Pseudomonas aeruginosa* [92].

Most studies focus on modifying the microorganism in order to improve the conversion of phytosterols into AD using genetic engineering or mutagens (chemicals and/or UV), as we can see in Figure 6. Besides, the screening of AD producers and their DNA sequencing is always in the spotlight since it helps to elucidate the metabolism processes of several microorganisms and establish a good genetic engineering strategy. Donova et al. [75] used chemical and UV irradiation mutagenesis, combined with a selection pressure by sitosterol, with the parental strain *Mycolicibacterium* sp. (basonym: *Mycobacterium* sp.) VKM Ac-1815D to identify those microorganisms producing AD and 9-OH-AD from sitosterol. The major AD producer was the strain *Mycolicibacterium* sp. VKM Ac-1815D, with a 73% molar yield. The selected mutant (*Mycolicibacterium* sp. 2–4 M) was able to produce 9-OH-AD as a major product (50% molar yield), in addition to AD (22% molar yield). Su et al. [81], for example, overexpressed the cyp125-3 gene in *Mycolicibacterium neoaurum* (basonym: *Mycobacterium neoaurum*) *TCCC 11978* to increase AD productivity, since this gene is directly involved in phytosterol degradation and the generation of NAD+ (which is needed for AD production).

Sixteen articles reported the use of strategies to reduce the degradation of the steroid nucleus by the study of the enzymes related to this effect (KstD and Ksh). Xu et al. [64] showed that by reducing the temperature from 37 to 30 °C, the degradation of the nucleus was decreased from 39.9 to 17.6%, possibly due to the inhibition of the putative activity of KstD and Ksh—key enzymes of the microbial catabolism of steroid compounds. In addition, they demonstrated that the degradation of phytosterols in *Mycolicibacterium* sp. (basonym: *Mycobacterium* sp.) follows the AD-ADD-′9-OH-ADD′ pathway. Wang et al. [106] proposed a two-stage bioprocess to reduce the degradation of the steroid nucleus in phytosterol bioconversion by *Mycolicibacterium neoaurum* (basonym: *Mycobacterium neoaurum*) NwIB-R10hsd4A; cell culture was performed at 30 °C (to reduce the oxidation of the nucleus) and bioconversion at 37 °C (to increase bioconversion enzyme activity), resulting in a reduction in core degradation to 17.6%.

Mechanisms to increase cell permeability were the subject of 4 articles [49,100,106,107]. Barry [123] reported that interference in the mycobacterial cell wall structure and physical architecture resulted in changes to its permeability. Korycka-Machała et al. [107] treated *Mycolicibacterium vaccae* (basonym: *Mycobacterium vaccae*) using ethambutol (EMB), an inhibitor of arabinogalactan-polysaccharide biosynthesis, to improve the performance of the intracellular degradation of β-sitosterol. EMB, in the presence of rifampicin (an antibiotic), resulted in greater AD accumulation than the control because it increased cellular sensitivity to antibiotics, consequently increasing cell permeability and AD productivity [106]. Rumijowska-Galewicz et al. [106] also used inhibitors of the biosynthesis of mycobacterial cell wall compounds (m-fluorophenylalania and D, l-norleucine) and observed an AD increase.

#### 3.2.2. Scientific Articles Categorized as “Process Improvement”—Micro Analysis

The manuscripts categorized as “process improvement”, i.e., improvements related to an increase in the biotransformation performance, were mostly related to the addition of chemical substances, culture medium modifications, and an increase in the solubility of the substrates (Figure 7).

Most works (40%) in the category “process improvement” are related to “chemical addition”, which means that AD production can be significantly increased by the addition of solvents, surfactants, cofactors, inducers, ionic liquids, etc. Yuan et al. [52] investigated the biocompatibility of 13 types of ionic liquids (LI) with resting cells of *Mycolicibacterium* sp. (basonym: *Mycobacterium* sp.), to produce AD from phytosterols and LIs with the anions [PF_6_] (hexafluorophosphate) and [NTf_2_] (bis(trifluoromethylsulfonyl)amide) were the most biocompatible. The biphasic system (LI/aqueous medium) containing [PrMIM] [PF_6_] (1-Propyl-3-methylimidazolium hexafluorophosphate) resulted in greater AD production (2.35 g·L^−1^) after 12 h.

The addition of chemical compounds was also performed to extract AD from the culture medium. Josefsen et al. [17] used ethyl acetate, while Thygs and Merz [108] performed a crossflow extraction technique with ethanol to extract the product. Huang et al. [4] added Amberlite XAD-7 resin to the culture medium as an adsorbent; subsequently, AD and ADD were purified using silica gel column chromatography and the effluent was then evaporated by depressurizing the system. Cruz et al. [98] addressed the use of di(2-ethylhexyl)phthalate (DEHP) as a reaction medium in the biotransformation of β-sitosterol into AD, with *Mycolicibacterium* sp. (basonym: *Mycobacterium* sp.) NRRL/B-3805 free suspended cells.

Some publications report the use of chemical agents to increase the biotransformation of phytosterols in 3,17-diketosteroides (AD, ADD, and 9-OH-AD). Josefsen et al. [17], for example, tested strategies to overcome the challenges of the insolubility of substrate (phytosterol) and product (AD, ADD) in water, such as the use of modified cyclodextrin (Me-β-CD and HP-β-CD). Indeed, 15% of publications discuss alternatives for increasing phytosterol solubility in the reaction medium using two-phase systems, while some of them are also related to chemical addition because the use of cyclodextrins and several types of organic solvents are proposed to obtain those systems. Xu et al. [100] evaluated 6 organic solvents and 8 cyclodextrins, to increase phytosterol solubilization in *Mycolicibacterium* sp. (basonym: *Mycobacterium* sp.) MB 3863 biotransformation. Ethanol and acetone were the best solvents and the methylated derivative of cyclodextrin (β-Me-CD) stood out. Pendharkar et al. [48] also verified several organic solvents with *Mycolicibacterium fortuitum* (basonym: *Mycobacterium fortuitum*) subsp. *fortuitum* NCIM 5239 and also concluded that ethanol is the best solvent for increasing phytosterol bioavailability for bacterial cells. The aqueous microdispersion technique of phytosterols with a particle size of 370 nm was studied by Mancilla et al. [83] by adding Tween 80 and sodium salts of sunflower fatty acids and with the use of a homogenizer.

Some studies related to medium composition evaluated the substrate type [100,104,105], describing the use of several phytosterols, either pure or directly from natural raw materials, such as soybean, canola, coconut, palm, corn, and even the unsaponifiable material of rice bran oil. Sripalakit and Saraphanchotiwitthaya [105] evaluated the feasibility and benefits of using phytosterols from vegetable oils and confirmed that canola oil resulted in the highest AD and ADD production compared to other vegetable oils.

Stefanov et al. [23] studied the effects of different process conditions (agitation speed, age and amount of inoculum, temperature, and additional carbon sources) on the biotransformation of phytosterols into AD and ADD by *Mycolicibacterium* sp. (basonym: *Mycobacterium* sp.) MB3683 in a two-phase system. A higher conversion (10–15%) was detected with the use of an inoculum cultivated for 16–20 h, at a temperature of between 34–35 °C, and agitation at 400 rpm. Media containing high concentrations of carbohydrates reduced bioconversion, due to the microbial preference for these types of organic carbon sources instead of phytosterols. Alternatives that increase oxygen transfer were studied by Su et al. [28]. Soybean oil (16%) increased the volumetric oxygen transfer coefficient (K_L_a) by 44% in *Mycolicibacterium neoaurum* (basonym: *Mycobacterium neoaurum*) TCCC 11979 culture, resulting in a peak of 55.76% AD molar yield.

#### 3.2.3. Scientific Articles Categorized as “Metabolic Intermediates”—Micro Analysis

The manuscripts categorized as “metabolic intermediates” were mainly related to phytosterol or cholesterol conversion to intermediates (84%). Only 6 documents reported the direct production of hormones by phytosterol [14,42,47,86,92,113]. The final step in the biosynthesis of testosterone, for example, is the reduction of AD to TS by 17β-hydroxysteroid dehydrogenase. Although the microbial transformation of sterols to AD is now utilized on an industrial scale, commercial production of TS from AD is still carried out using chemical synthesis [86]. These studies show the possibility of transforming phytosterols directly into hormones, which would be an advantage for industrial production. ADD and 9-OH-AD are also important intermediates for the synthesis of hormone pharmaceuticals [15,20]. Therefore, most studies focus on those products, and the same strategies used to increase AD production are also evaluated in those studies [15,20,21,32,33,37,39,40,51].

The studies related to direct hormone production from phytosterols report, for example, the development of a method for the efficient production of dehydroepiandrosterone from phytosterols in a vegetable oil/aqueous two-phase system by *Mycolicibacterium* sp. (basonym: *Mycobacterium* sp.) [42]. The biotransformation in a 30-L stirred bioreactor, with 25 g L^−1^ of substrate, produced 16.33 g L^−1^ of DHEA after 7 days. The testosterone production was performed by Fernández-Cabezón et al. [47] to make the model strain *Mycolicibacterium smegmatis* (basonym: *Mycobacterium smegmatis*) suitable for hormone production, to compete with the current chemical synthesis method by using a genetic engineering strategy.

### 3.3. Patents General Aspects—Macro Analysis

Figure 8 shows the number of patents that report the biotransformation or biosynthesis of AD between 1976 and March 2021. A low number of patents between 1977 and 1980 is evident, while an increase in that number is observed between 1981 and 1982. However, after that period, only 5 patents were registered over the years.

The few granted patents in the first four years [124,125,126,127,128,129] focus on obtaining molecules through the microbial transformation of steroids having 17-alkyl side chains of from 2 to 10 carbon atoms, such as AD and ADD.

After 1980, patents have reported AD as a product but no longer as a substrate, as in previous years. In a total of 17 documents, the Upjohn Company appears as the patent holder of 12 granted patents [127,130,131,132,133,134,135,136,137,138,139,140]. These 12 documents discuss the different substrates and microorganisms used for AD production or even with ADD production as a by-product, as well as the separation methods of both.

The number of granted patents peaked in 1982. The 8 documents published over this year reported the different process conditions and mutant microorganisms used in the transformation of different substrates into AD or AD/ADD mixtures. Patent number 4,345,030 [137], for example, mentions different substrates, such as sitosterol, campesterol, stigmasterol, and cholesterol, as well as different genera of microorganisms used in the mutant strain, such as *Arthrobacter*, *Bacillus*, *Brevibacterium*, *Corynebacterium*, *Microbacterium*, *Nocardia*, *Protaminobacter*, *Serratia* and *Streptomyces*.

From 1982 until now, only five new documents were registered in the USPTO database. In 1995, patent number 5,418,145 [122] was granted to Schering Aktiengesellschaft, a German pharmaceutical company. This publication reports AD production by fermenting alpha-sitosterol with microorganisms capable of the side-chain degradation of sterols. Patent number 7,241,589 [121] was granted to the Korean pharmaceutical company Eugene Science, Inc. in 2007. The document is related to the high conversion of cholesterol and emulsified cyclodextrin-cholesterol into an AD/ADD mixture. The inventors are the same as those in patent application 20,040,152,153 [121]. However, this application was filed by the actual inventors and not by a company. This patent [121], granted in 2004, describes AD/ADD production using different substrates and distinct process conditions. The two other patent applications, filed in 2013 and 2015, report the use of AD and ADD in pharmaceutical compounds that act as aromatase enzyme inhibitors and in the treatment of endocrine diseases, hormonal disorders, cancer, and other estrogen-related pathologies.

The United States is the largest patent-holding country, with more than 90% of the total granted patents. The North American emphasis can be attributed, among other factors, to the large number of documents granted to the pharmaceutical company Upjohn, founded in 1886 in Kalamazoo, Michigan. The only two granted patents that do not belong to the United States were granted to Schering Aktiengesellschaft in 1995 and Eugene Science, Inc. (Eugene, Oregon) in 2007.

Concerning the type of patent-holding institution, all granted patents belong to companies (Figure 9). However, not all patent applications were filed by companies. One of them was filed by its own inventors, and the two final applications were filed by Emory University, in collaboration with Hauptman Woodward Medical Research Institute and the Research Foundation for the State University of New York. The Hauptman–Woodward Medical Research Institute is a research center that works to discover the causes and treatment of several diseases, usually collaborating with other institutions and universities.

Among the 18 patents granted to the Upjohn Company, 8 of them discuss the possibility of using AD as a substrate for mutant microorganisms to transform it into other steroids, while the other documents aim at producing AD from different substrates and mutant microorganisms. Therefore, 12 documents (including that granted to Eugene Science Inc.) have relevant information about AD biosynthesis, detailed in the micro perspective.

According to the American Chemical Society (ACS), the Upjohn Company was the world’s leading producer of steroid intermediates and drugs in 1990. The company’s research and development department innovated medicinal chemistry and microbial and chemical transformations in its steroid drug manufacturing processes. The company joined with Pharmacia AB to create Pharmacia and Upjohn (Peapack, NJ, USA), which currently belongs to Pfizer.

### 3.4. Categorizing the Patents by Groups—Meso Analysis

Approximately 60% of the documents focus on technologies and the routes to produce AD, while almost 40% discuss the use of AD in the production process of other compounds. Concerning the granted patents only, those percentages are about 65% and 35%, respectively. Table 2 shows the process conditions for AD production reported in most patents.

The documents 7,241,589 [121], 5,418,145 [122]; 4,358,538 [140], 4,345,034 [139], 4,345,033 [138], 4,345,030 [137], 4,345,029 [136], 4,339,539 [120], 4,328,315 [135], 4,304,860 [134], 4,293,646 [133], 4,293,645 [131] and 4,293,644 [132] were categorized as “Technologies/Routes”. As previously mentioned, the abstract, claims and description of every patent were analyzed. Those patents describe different methods of obtaining AD and, in some cases, ADD also, by degrading phytosterols, cholesterol and other compounds using mutant microorganisms. The approach involved the different mutant microorganisms, different process conditions, and different substrates used in the processes, in addition to detailing the modifications in microorganisms.

The documents 4,039,381 [124], 4,042,459 [125], 4,097,335 [126], 4,098,647 [127], 4,176,123 [128], 4,329,432 [129] and 4,211,841 [130] were categorized as “use of a molecule”. By analyzing the abstract, the claims, and the descriptions of these patents, there are several examples of the use of AD as a substrate to produce compounds using mutant microorganisms. These documents describe the use of AD, for example, in a mixture of steroids that consists of AD, sitosterol, cholesterol, stigmasterol, and campesterol, among others. The documents also contain information about the mutant microorganisms and compare the results obtained by varying the substrates and the microorganisms used.

The patents categorized as “technologies/routes” have only appeared since 1981, which shows an interest in this molecule’s production in this period. Between 1981 and 1982, 12 patents were granted to the Upjohn Company, and 11 patents were classified as “technologies/routes”, showing a real effort to develop an AD production process by this company. In addition, the only two patents granted after 1982 were categorized as “technologies/routes” and were granted to Schering Aktiengesellschaft in 1995 and Eugene Science, Inc. in 2007. There are three other documents published after 1982. Two of them were categorized as the “use of a molecule” and were filed in 2013 and 2015 by a research center, in collaboration with a university and a state foundation; they do not describe the industrial production of this molecule since they focus on the use of AD. The other was filed in 2004 by the same inventors as the patent granted to Eugene Science, Inc. in 2007. Note that this patent application has not been granted so far, which means it will probably not be granted, since 14 years have passed.

### 3.5. Granted Patents Categorized as “Technology/Routes”—Micro Analysis

The granted patents categorized as “technology/routes” were analyzed according to three perspectives: “microorganisms”, “substrates” and “process conditions”.

All granted patents report the use of mycobacteria in the process of the biotransformation of sterols into AD. *Mycolicibacterium fortuitum* (basonym: *Mycobacterium fortuitum*) NRRL B-11045 appears in three documents, followed by *Mycolicibacterium fortuitum* NRRL B-8153, *Mycolicibacterium fortuitum* NRRL B-8154 and *Mycolicibacterium fortuitum* NRRL B-11359, which appear in 2 documents each. The strains *Mycolicibacterium fortuitum* NRRL B-11358, *Mycolicibacterium fortuitum* NRRL B-8128, *Mycolicibacterium* sp. (basonym: *Mycobacterium* sp.) NRRL B-3805, *Mycobacterium spec*. NRRL B-3683, and *Mycolicibacterium fortuitum* EUG-119 (KCCM-10259) were used in only one granted patent. The last-mentioned strain was the same as that used in the patent application filed in 2004 by the same inventors as the patent granted to Eugene Science, Inc. in 2007.

All these mutants present high AD production performance. In the case of the patents granted to the Upjohn Company, the mycobacteria used are adaptive mutants. For example, in the patent 4,345,034 [139], *Mycolicibacterium fortuitum* (basonym: *Mycobacterium fortuitum*) NRRL B-11359 was obtained from *M. fortuitum* NRRL B-11358, which is an adaptive mutant of *Mycolicibacterium fortuitum* NRRL B-11045. In all cases, the selected microorganism was able to transform steroids with 17-alkyl side chains of from 2 to 10 carbon atoms into AD as the main product.

The substrates that can be used in AD production are sitosterol, cholesterol, stigmasterol, campesterol, and a cyclodextrin–cholesterol complex. The first four mentioned substrates appear in all 11 patents granted to the Upjohn Company, while the cyclodextrin-cholesterol complex only appears in the patent granted to Eugene Science, Inc. in 2007. The patent granted to Schering Aktiengesellschaft reports the use of sitosterol only. All the Upjohn Company’s patents describe AD production from sitosterol and suggest the replacement of this substrate by cholesterol, stigmasterol, campesterol, or a mixture of them, obtaining AD as the main product. However, the Eugene Science, Inc. patent mentions the use of cholesterol and the cyclodextrin–cholesterol complex as substrates for AD production.

The Eugene Science, Inc. patent reports that the cyclodextrin-cholesterol complex, which can be extracted from milk by the addition of β-cyclodextrin, is an excellent low-cost substrate for producing AD by microorganisms. It can be easily dissolved and dispersed in an aqueous culture medium. The high interaction of cyclodextrin with steroids enables the cyclodextrin–sterol complex preparation, which increases the productivity of AD. The AD/ADD ratios obtained using this complex are about 2.3 greater than with the conventional method using only emulsified cholesterol. The document also points out that sterols such as cholesterol, sitosterol, campesterol, stigmasterol, and ergosterol can be used in cyclodextrin–sterol complex production. The patent highlights the high performance and easy preparation of the cyclodextrin–sterol complex as important issues for its industrial application.

All the granted patents analyzed in this work describe AD production on a laboratory scale. The Upjohn Company’s patents report that the fermentation temperature can vary from 25 °C to 37 °C, with 30 °C being used in all documents to optimize the microorganism growth and the conversion of the substrate into the final product. AD production was also performed at 30 °C in the processes described by Eugene Science, Inc. and Schering Aktiengesellschaft, which means that 100% of the granted patents report the same fermentation temperature.

On the other hand, there is a substantial difference in fermentation time. The Schering Aktiengesellschaft patent describes a 96-hour process, while Eugene Science Inc. reports 5 days, and the Upjohn Company, 14 days to produce AD. The shortest fermentation time reported by Eugene Science Inc. and Schering Aktiengesellschaft may be related to the substrates used, which were both emulsified. However, the microorganism used by Eugene Science, Inc. is also different from all those used by the Upjohn Company.

Regarding the agitation of the system, all the patents mentioned its importance; nevertheless, only Eugene Science, Inc. and Schering Aktiengesellschaft provided the rotation used, 200 and 220 rpm, respectively. The agitation is important to promote microorganism growth and increase the efficiency of the process.

## 4. Challenges, Opportunities, and Development Efforts

According to Market Watch [141], the Global AD Market was valued at USD 190 million in 2018 and is estimated to reach USD 210 million by 2025. The major uses of AD include fertility control, arthritis, and infectious inflammation pharmaceuticals. However, arousal and sexual function increase, and enhanced recovery and performance during exercise have driven the AD market. The growing fitness market and interest in higher testosterone levels have encouraged the AD market. To achieve stringent fitness targets, athletes and fitness enthusiasts will consume androstenedione products [119,141].

The technological prospecting of scientific articles in the SCOPUS, Web of Science, and Google Scholar databases over the years 2000 to 2021, and of the patents in the USPTO (the United States Patent and Trademark Office) database from 1976 to 2021, regarding androstenedione biosynthesis from phytosterol, provided a holistic view of research and innovation trends. Based on the distribution of articles over the years, a peak in publications in 2017 was detected, evidencing a possible resumption of interest in the subject. The countries that most contributed in relation to the articles were China, India, Russia, the United States, and Spain. In the case of patents, the United States is the leading country.

Articles and patents show some similarities in relation to the topics addressed, reinforcing the trends, characteristics, and particularities of the AD biosynthesis process. Scientific articles aimed at process optimization could be related to the physical or chemical aspects of the process or the microbial agent and not to the discovery of new routes for AD production from phytosterols. The most discussed subjects within this context were the use of genetic engineering and mutagens, chemical addition, and alternatives to increase phytosterol solubility. The genetic modification studies aimed to reduce byproduct formation and increase AD yield. Many of the chemicals added to the culture medium were, in fact, related to reducing substrate and product insolubility, mostly ionic liquids, surfactants, cyclodextrins, and solvents. In general, the patents propose methodologies to optimize AD production, evaluating the influence of different *Mycolicibacterium* (basonym: *Mycobacterium*) mutants and the sterols being used as substrates (cholesterol and phytosterols).

Compared to the granted patents, the scientific articles of AD biosynthesis are more innovative in terms of methodologies and technologies. This is not a common trend, since patents are usually a result of several previous scientific studies and, therefore, they are the outcome of all knowledge on the subject applied in the development of a marketable technology. However, the patents analyzed herein predate the scientific articles, which means that when these studies were carried out, all information present in these patents was already known. According to USPTO, there are only two patents focused on technologies for AD biosynthesis after 2000. One of them was granted to Eugene Science, Inc. and describes a promising innovation, the use of a cyclodextrin–cholesterol complex as a substrate in the biosynthesis of AD.

Table 2 brings together the main aspects of AD and ADD biosynthesis: incubation temperatures can vary from 29 °C to 37 °C; it can last from 72 h to 14 days; it is agitated at 140 to 220 rpm; and vegetable oils, mainly soybean, can be used as the source of a mixture of phytosterols. Additionally, genetic modifications and ionic liquid or cyclodextrin addition are the typical strategies to significantly increase the efficiency of the biotransformation process.

The results obtained through the analyses involved in this technological prospecting study are of paramount importance for planning and developing new processes and technologies to produce steroid intermediates and suggest great advances in the biosynthesis of androstenedione from phytosterols.

## Figures and Tables

**Figure 1 molecules-27-03164-f001:**
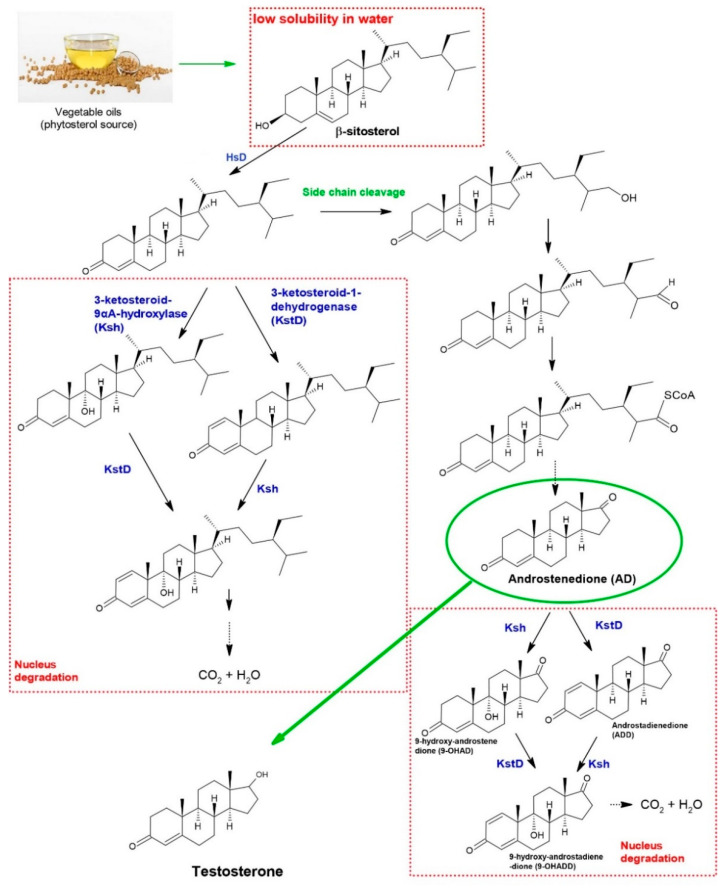
Biochemical pathway of androstenedione (AD) production from phytosterol, obtained from vegetable oils—mainly β-sitosterol. Dashed arrows indicate multiple enzymatic steps. HsD: 3β-hydroxysteroid dehydrogenase. The red dashed borders indicate the obstacles to AD production: the substrate’s low solubility and steroidal nucleus degradation. Green arrows indicate chemical reactions or extraction procedures [1,2,3,6].

**Figure 2 molecules-27-03164-f002:**
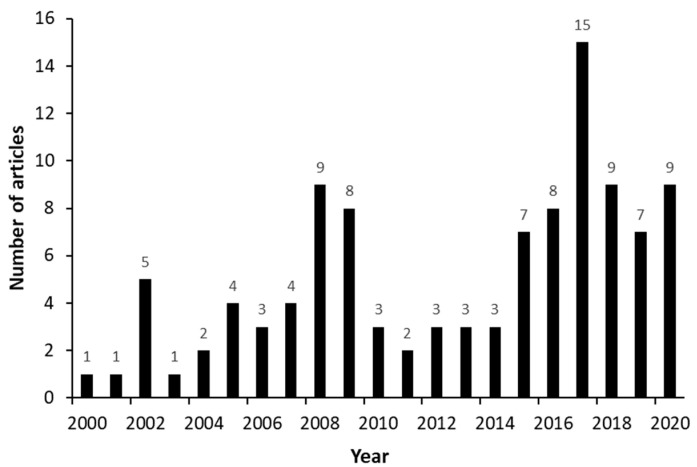
Numbers of publications related to androstenedione (AD) production from phytosterols over the years (2000 to 2021).

**Figure 3 molecules-27-03164-f003:**
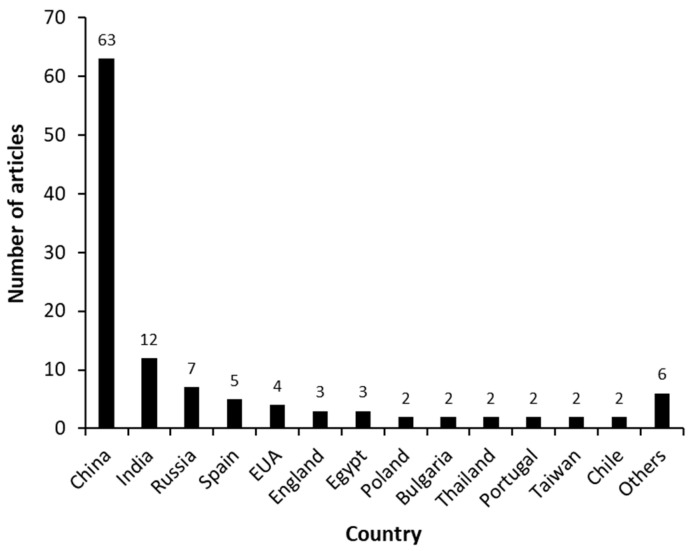
Scientific articles that report the biosynthesis of androstenedione (AD) from phytosterols between 2000 and 2021, shown by country.

**Figure 4 molecules-27-03164-f004:**
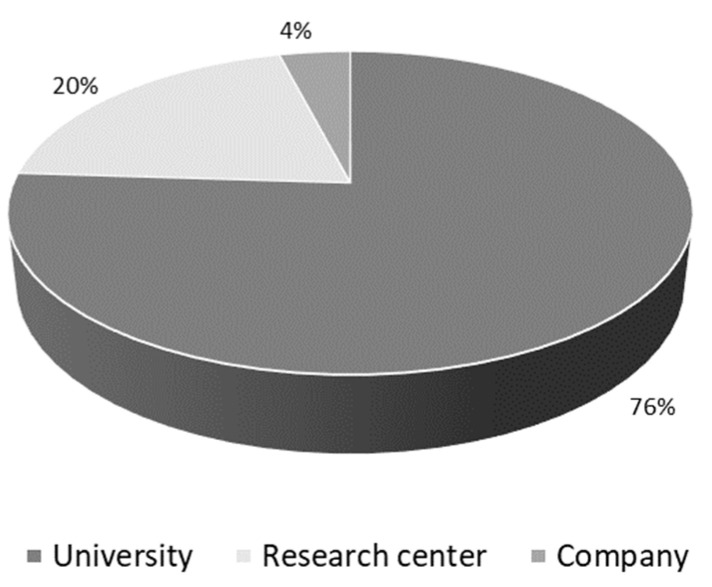
Publications about the androstenedione (AD) biosynthesis from phytosterols by different institutions between 2000 and 2021.

**Figure 5 molecules-27-03164-f005:**
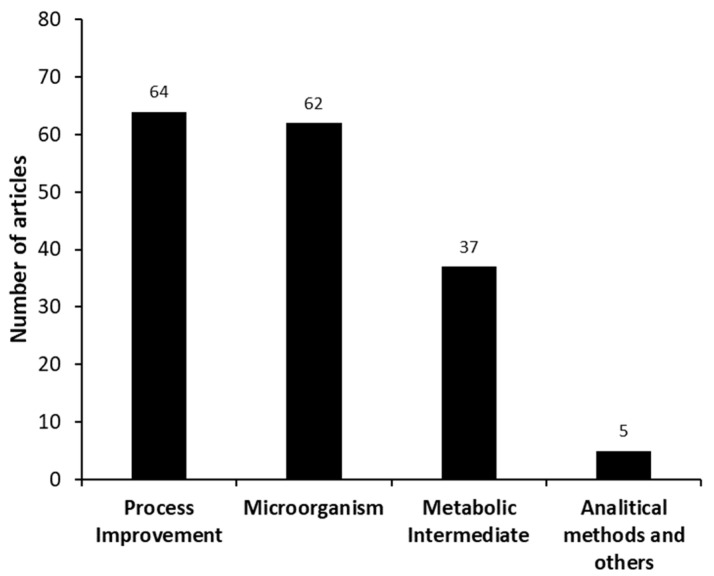
Different categories identified in the set of manuscripts involving androstenedione (AD) production from phytosterols, as found in the systematic search (meso analysis).

**Figure 6 molecules-27-03164-f006:**
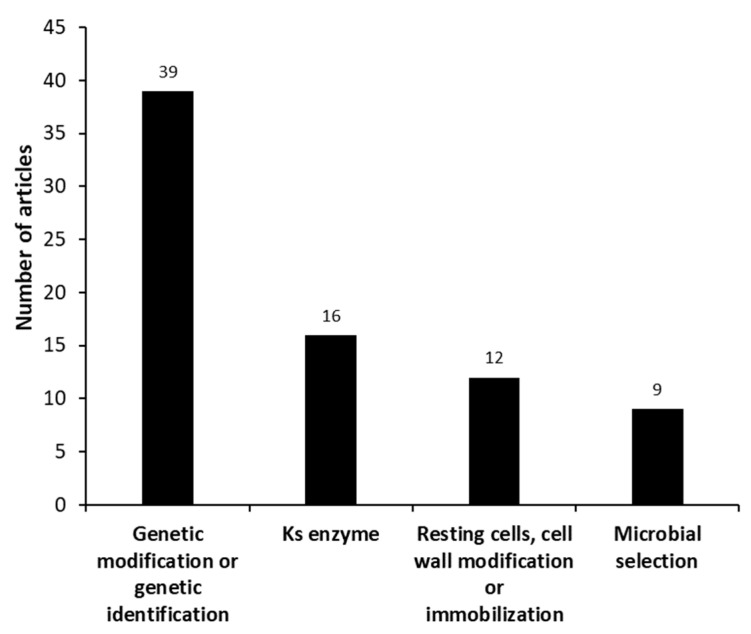
Different groups identified in the set of manuscripts, categorized as “microorganism” (microanalysis), for microbial androstenedione production from phytosterols.

**Figure 7 molecules-27-03164-f007:**
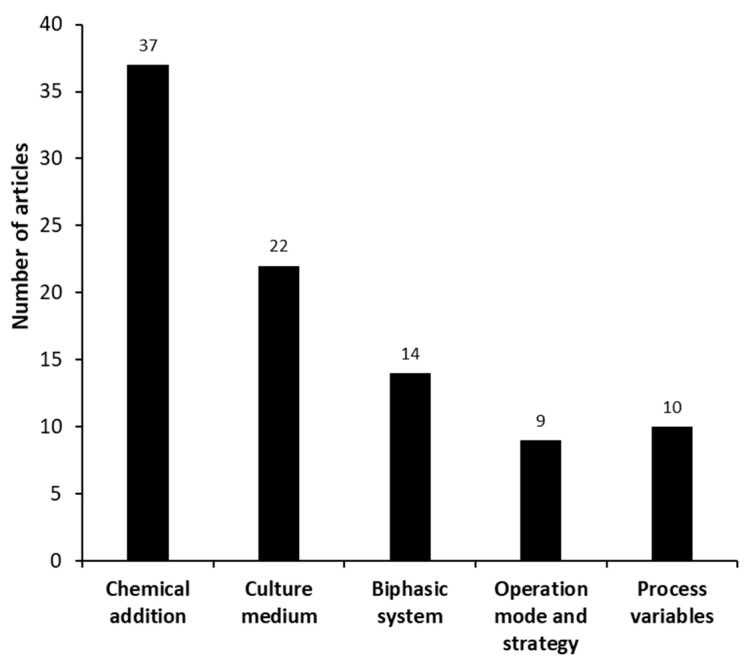
Different groups identified in the set of manuscripts categorized as “process improvement” (microanalysis) for microbial androstenedione production from phytosterols.

**Figure 8 molecules-27-03164-f008:**
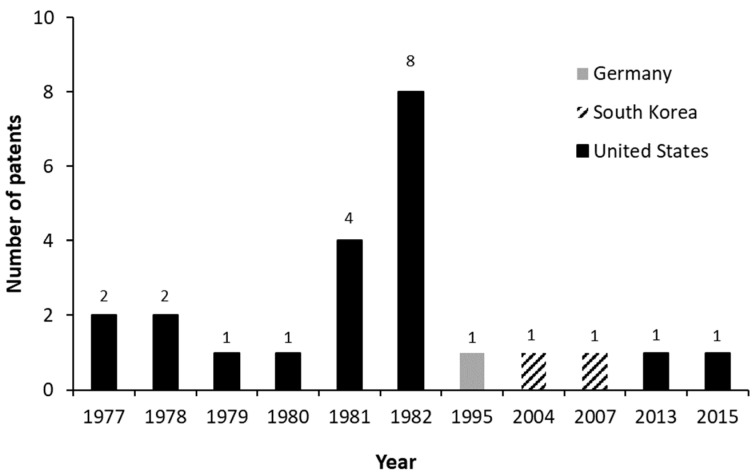
Patents that report the biotransformation or biosynthesis of androstenedione (AD) between 1976 and March 2021.

**Figure 9 molecules-27-03164-f009:**
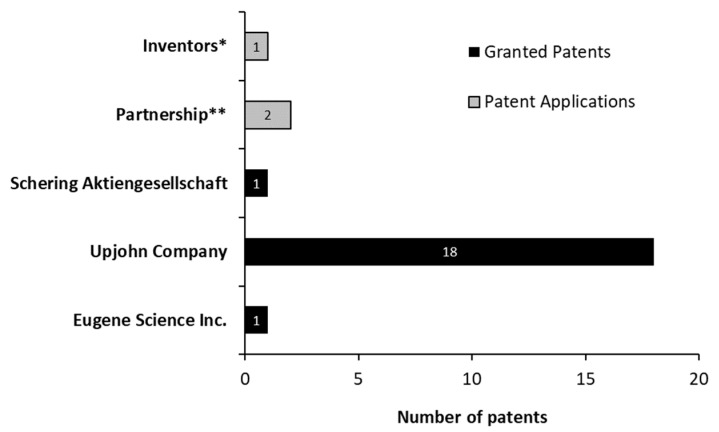
Patent-holding institutions and patent applications for androstenedione production between 1976 and March 2021. * Filled by Noh, Seung-Kwon; Kim, Myung-Kuk; Yoon, Won-Tae; Park, Kyung-Moon; Park, Sang-Ok. ** Partnership between Emory University, Hauptman–Woodward Medical Research Institute, and the Research Foundation for the State University of New York.

**Table 1 molecules-27-03164-t001:** Categories of the meso and micro perspectives of the technological prospecting study.

Meso Perspective Group	Micro Perspective Group	References
Microorganism	Genetic modification or genetic identification	[4,6,10,13,14,15,18,21,22,29,30,31,32,39,43,45,46,47,51,53,55,56,57,60,61,62,65,74,75,81,85,86,89,90,92,93,95,110,113]
	Ks enzyme *	[6,10,13,14,15,29,35,43,53,55,57,62,87,93,97,104]
	Resting cells, cell wall modifications or immobilization	[11,19,34,38,49,52,54,58,100,106,107,109]
	Microbial selection	[24,37,70,71,89,91,96,101,109]
Process Improvement	Chemical addition	[11,17,18,20,22,23,27,28,33,38,39,40,44,48,49,52,59,63,69,72,73,78,80,82,83,84,94,98,99,100,102,103,106,107,111,112,114]
	Culture medium	[11,22,23,25,34,36,44,54,60,66,68,70,71,77,80,82,92,105,110,112,113,115]
	Biphasic system	[19,23,27,42,44,66,68,73,78,82,83,94,99,114]
	Operational mode or strategy	[13,14,16,44,45,50,79,101,104]
	Process variables	[22,23,36,54,59,64,80,86,112,113]
Metabolic Intermediates and Hormones	PS ** or Co *** conversion into intermediates	[6,15,19,20,21,22,25,26,32,33,37,39,40,41,50,55,61,65,69,71,72,73,74,75,79,84,95,96,97,101]
	Hormone production from PS **	[14,42,47,51,86,92,113]
Analytical Methods and others	Analytical methods	[12,67,88]
	Others	[76,108]

* Ks enzyme: The enzymes 3-ketosteroid-9α-hydroxylase (Ksh) and 3-ketosteroid-1-dehydrogenase (KstD); ** PS: phytosterol; *** Co: cholesterol.

**Table 2 molecules-27-03164-t002:** Process variables for microbial androstenedione (AD) and androstadienedione (ADD) production, from the most relevant scientific articles and patents found in a systematic search.

Microorganism ^1^	Substrate	Genetic Modifications	Reactional Conditions	Results	Differential	Ref.
**Scientific Articles**						
*Mycolicibacterium**neoaurum* TCCC 11978 (MNR M3)	Sterol mixture, weight percentage: 51.7% β-sitosterol, 27.2%; stigmasterol, 17.1% campesterol, and 4.0% brassicasterol—Soybean oil	Cofactor engineering: modification of enzymes related to NADH * and NAD^+^ * metabolism	pH: 7.2; 30 °C; 140 rpm; 144 h	conversion ratio 94%	nicotinic acid in the phytosterols fermentation system to increase intracellular NAD^+^/NADH	[28]
*Mycolicibacterium neoaurum* TCCC 11979	Phytosterol (98.4% purity)	-	29 °C; 140 rpm; 120 h	molar yield of AD 55.8%	Oxygen vectors (n-hexadecane, perfluorohexane, soybean oil, PDMS, and PMPS *)	[59]
*Mycolicibacterium**neoaurum* TCCC 11978 (MNR M3)	Sterol mixture, weight percentage: 51.7% β-sitosterol, 27.2% stigmasterol, 17.1% campesterol, and 4.0% brassicasterol	Overexpression of cytochrome p450 125 (cyp125-3)	pH: 7.2; 30 °C; 140 rpm; 120 h	Conversion: 96%; 1.98 g·L^−1^ in 96 h	phytosterols (3 g·L^−1^) and HP-β-CD ** (25 mM)	[81]
*Mycolicibacterium**neoaurum* TCC 11028 (MNR M3)	Phytosterol (98.4% purity/3 g·L^−1^)	Overexpression of nicotinic acid phosphoribosyltransferase (NAPRTase)	pH 7.2; 29 °C; 200 rpm; 96 h	molar yield of AD (D) (94.9%)	HP-CD ** (0 or 25 mM)	[90]
*Mycolicibacterium neoaurum* NwIB-R10hsd4A	Phytosterol	-	T_1_ 30 °C; T_2_ 37 °C	24.7 g·L^−1^	two-step bioprocess, cell culture at 30 °C and bioconversion with resting cells at 37 °C	[104]
*Mycolicibacterium* sp. VKM Ac-1817D	Phytosterol	-	30 °C; 200 rpm	11 mmol/L; 0.3 mmol/h/g dry cell	MCD ****	[39]
*Mycolicibacterium* sp.	4.5% β-sitosterol; 26.4% campesterol; 17.7% stigmasterol; 3.6% brassicasterol	Deletion *kshA1* and *kstD1* *****	pH = 8; T = 37 °C; 200 rpm; 72 h	AD, 3.1 g·L^−1^	HP-β-CD ***; increase in culture temperature to 37 °C to reduce nucleus degradation	[64]
*Mycolicibacterium neoaurum* NwIB-01	Soybean phytosterols	Inactivation and augmentation of the primary 3-Ketosteroid-δ1-Dehydrogenase	30 °C; 300 rpm; airflow 0.5 vvm; 96 h	ADD, 4.23 g·L^−1^; AD, 1.76 g·L^−1^; (57.8% mole conversion)	-	[62]
*Mycolicibacterium* sp. MB 3683	Phytosterol	-	30 °C; 200 rpm; 30 h	1.3–1.4 g·L^−1^	Cholinium; amino acids Ionic liquids; Best: 1% (*v*/*v*) [Ch][Asp]	[38]
*Mycolicibacterium neoaurum* TCCC 11978 C2	51.7% sitosterol; 27.2% stigmasterol; 17.1% campesterol; 4.0% brassicasterol	-	30 °C; 140 rpm; 120 h	84.8% mole conversion	HP-β-CD ***	[102]
*Mycolicibacterium neoaurum*, *Pimelobacter simplex,* and *Rhodococcus erythropolis*	soybean sterols (20–30 g/L)	-	30 °C; 220 rpm; phytosterol load of 30 g/L over 144 h	AD: 14.5–15.2 g·L^−1^	Mixture of soy steroids (20–30 g/L) in the form of small crystals in suspension (particle size 5–15 μm)	[33]
*Mycolicibacterium* sp. DSM-2967	phytosterol-containing vegetable oils	-	pH 7.8; at room temp.; 200 rpm	Best: with canola oil; yield: 7.92 mg/100 mL	Phytosterol-containing vegetable oils directly converted to AD	[105]
*Mycolicibacterium* sp. MB 3683	Phytosterols	-	Ionic liquid addition at 84 h, 20:1 (*v*/*v*, aqueous/IL),	AD production reached 2.23 g·L^−1^ after 5 days	Ionic liquid to increase low substrate solubility	[114]
*Alkalibacterium olivoapovliticus*	olive oil	-	72 h; 30 °C	Conversion: 90%	Concrete was used as a tool to immobilize the microorganism	[109]
*Moraxella ovis*	Rice bran oil (RBO)	-	36 h, pH 7; 30 °C	0.22 mg AD/40 mg RBO	The unsaponifiable matter of rice bran oil was used as a raw material	[71]
*Mycolicibacterium* sp. B-3805S/*Mycolicibacterium* sp. NRRL B-3683	Phytosterol	nitrosoguanidine (NTG) mutagenesis	5-L surface-aeration microprocessor-controlled fermentor; 30 °C	Conversion: 70.6%	-	[4]
**Patents**						
*Mycolicibacterium phlei* NRRL B-8154	Sitosterol, cholesterol, stigmasterol and campesterol	Nitrosoguanidine mutagenesis	30 °C; 14 days	-	-	[120]
*Mycolicibacterium fortuitum* EUG-119 (KCCM-10259)	Cyclodextrin-sterol complex	-	30 °C; 5 days; 200 rpm	-	-	[121]
*Mycolicibacterium* sp. NRRL B-3805	Alpha-sitosterol (AS)	-	30 °C; 4 days; 220 rpm	160 mg AD/1000 mg AS	-	[122]

^1^ The microbial species names used are the ones re-classified by Gupta et al. [120] but that are mentioned in the cited manuscripts as the original names: *Mycolicibacterium* sp. (mentioned as *Mycobacterium* sp.); *Mycolicibacterium neoaurum* (mentioned as *Mycobacterium neoaurum*); *Mycolicibacterium fortuitum* (mentioned as *Mycobacterium fortuitum*); *Mycolicibacterium phlei* (mentioned as *Mycobacterium phlei*). * NADH and NAD^+^: Nicotinamide adenine dinucleotide (reduced and oxidized, respectively). ** PDMS polydimethylsiloxane and PMPS polymethylphenylsiloxane. *** HP-β-CD hydroxypropyl-β-cyclodextrin. **** MCD methylated β-cyclodextrin. ***** *kshA1* and *kstD1*: Genes involved in the expression of the enzymes 3-ketosteroid-9α-hydroxylase (Ksh) and 3-ketosteroid-1-dehydrogenase (KstD), respectively.

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
