# Peer review of "Biotransformation of Phytosterols into Androstenedione—A Technological Prospecting Study"

_molecules, 2022, doi:10.3390/molecules27103164_

Round 1

Reviewer 1 Report

The full review is on file. My comments and suggestions are worth considering, but please focus on points 2, 5, and 10. In my opinion, they need refinement to publish this work. 

Reviewer 2 Report

Please see the attached file-Comments for molecules-1695086.

Reviewer 3 Report

The review is on microbial production of androstenedione from sterols. It should be noted that many comprehensive reviewes have been published during the past decade that deeply analyze the current state and prospects in this field and cover different aspects of phytosterol bioconversion to C19-steroids including androstenedione (see, for example, Zhao et al., Biotech Adv DOI:10.1016/j.biotechadv.2021.107860 and others). Unfortunately, this review is not in a scope of the journal "Molecules" and the section "Applied Chemistry". Besides, this review does not contain an in-depth analysis of the scientific and patent literature and does not reflect current trends and prospects, as well as recent achievements in the field of microbial production of androstenedione.

Round 2

Reviewer 2 Report

The manuscript has been greatly improved by the authors.

Reviewer 3 Report

The revised version of the manuscript has been improved, but cannot be recommended for the publication in its present state.

Comments:

1) It should be noted that currently, the main AD-producing strains that previously named as Mycobacterium (e.g. Mycobacterium neoaurum, M. smegmatis, etc.) are re-classified as Mycolicibacterium. The previous genus Mycobacterium has been divided on 5 different genera, and currently the genus Mycobacterium contains the pathogenic species such as Mycobacterium tuberculosis, while the soil-origin fast-growing species (such as above-mentioned AD-producers) relate now to Mycolicibacterium. For reference, use Gupta et al., 2018, Frontiers in Microbiology, doi:10.3389/fmicb.2018.00067; Oren and Garrity, 2018, Int J Syst Evol Microbiol doi:10.1099/ijsem.0.002711. This important information shall be added to the review, and current names of the microorganisms shall be given along with previous (at least, when mention firstly).

2) When indicating the name of the organism which species is not defined, pls do not italicize "sp." Pls correct it both in the Abstract (line 32) and further the whole text

3) Ref.9 in the list of references does not correspond to Zhao et al. (line 84). The corresponding reference (Zhao et al.) is ommitted in the list of references.

4) In many places in the text and in the list of references the names of microorganisms are not italicized. Pls correct

5) Line 45. Full chemical name of androstenedione and its corresponding CAS No. shall be given

6) Line 51. It is well established that 3b-ol-5-ene to 3-keto-4-ene modification is mainly catalyzed by 3b-hydroxysteroid dehyrogenases, and the key role of cholesterol oxidases is not proven for AD-producing strains and other sterol-transforming actinobacteria (for reference, see a review by Kreit, 2017, doi.org/10.1093/femsle.fnx007, or Ivashina et al., J Steroid Biochem Mol Biol, 2012 doi.org/10.1016/j.jsbmb.2011.09.008). The corresponding corrections shall be done in the text and in Fig.1.

7) The authors write (Line 56-57):"sitosterol branched side-chain cleavage requires cofactor regeneration." Why sitosterol only? Many steps in the sterol degradation pathway requires cofactor regeneration.

8) Besides, describing of only sitosterol in this paragraph (Line 56 and further) is looking strange. It is better to describe degradation of phytosterols because phytosterols are mainly used as starting materials for AD production.

9) Phrase "nucleic degradation of AD in ADD or 9-OH-AD" is not correct. In this case is better to write:"degradation of steroid core". Pls note that ADD and 9-OH-AD are formed not by "nucleic degradation of AD" as you indicated.

10) Line 266. Misprint in a word "Nocardia" (not Norcadia)

11) Line 350. "diketosteroids" (not diketoesteroids)

12) Line 386. The enzyme accounting for this reaction is 17b-hydroxysteroid dehydrogenase (not 17-keto reductase)

13) Line 450. Pls correct: 1980 (not 980)

14) Line 423 "different genera" (not "a different genus")
